# Preoptic leptin signaling modulates energy balance independent of body temperature regulation

**Sangho Yu[1]\*, Helia Cheng[1], Marie François[1], Emily Qualls-Creekmore[1], Clara Huesing[1], Yanlin He[2], Yanyan Jiang[3], Hong Gao[3], Yong Xu[2,4], Andrea Zsombok[3], Andrei V Derbenev[3], Eduardo A Nillni[4,5], David H Burk[1], Christopher D Morrison[1], Hans-Rudolf Berthoud[1], Heike Münzberg[1]\***

[1]Neurobiology of Nutrition and Metabolism Department, Pennington Biomedical Research Center, Louisiana State University System, Baton Rouge, United States; [2]Children's Nutrition Research Center, Department of Pediatrics, Baylor College of Medicine, Houston, United States; [3]Department of Physiology, School of Medicine, Tulane University, New Orleans, United States; [4]Department of Molecular and Cellular Biology, Baylor College of Medicine, Houston, United States; [5]The Warren Alpert Medical School, Department of Medicine, Molecular Biology, Cell Biology and Biochemistry, Brown University, Providence, United States

\*For correspondence:
sangho.yu@pbrc.edu (SY);
Heike.Munzberg@pbrc.edu (HM)

**Competing interests:** The authors declare that no competing interests exist.

**Abstract** The adipokine leptin acts on the brain to regulate energy balance but specific functions in many brain areas remain poorly understood. Among these, the preoptic area (POA) is well known to regulate core body temperature by controlling brown fat thermogenesis, and we have previously shown that glutamatergic, long-form leptin receptor (Lepr)-expressing neurons in the POA are stimulated by warm ambient temperature and suppress energy expenditure and food intake. Here we further investigate the role of POA leptin signaling in body weight regulation and its relationship to body temperature regulation in mice. We show that POA Lepr signaling modulates energy expenditure in response to internal energy state, and thus contributes to body weight homeostasis. However, POA leptin signaling is not involved in ambient temperature-dependent metabolic adaptations. Our study reveals a novel cell population through which leptin regulates body weight.
DOI: https://doi.org/10.7554/eLife.33505.001

## Introduction

The POA coordinates multiple autonomic and behavioral responses that are critical for survival (*McKinley et al., 2015*). Notably, the POA controls sympathetic nerve activity (SNA) to brown adipose tissue (BAT) and modulates energy expenditure to defend core body temperature in response to ambient temperature changes (*Morrison et al., 2014*). In addition, the POA was further implicated in food intake modulation under a changing thermal environment (*Brobeck, 1960*; *Andersson and Larsson, 1961*). For example, rats with POA lesions had difficulty not only maintaining core temperature but also adjusting food intake upon ambient temperature changes (*Hamilton, 1963*; *Hamilton and Brobeck, 1964*). Despite its prominent role in regulating energy expenditure and implication in controlling food intake, the POA has not been investigated as an important brain area for body weight regulation (*Yu et al., 2018*).

We have previously shown that POA Lepr neurons (Lepr[POA] neurons) are stimulated by warm ambient temperature and mediate warm-adaptive responses, especially by decreasing both energy

expenditure and food intake (*Yu et al., 2016*). Importantly, chronic chemogenetic stimulation of these neurons in mice resulted in body weight loss caused by reduced food intake that surpassed the decrease in energy expenditure. This result proposes that the manipulation of POA neural circuits can affect body weight. Another Lepr-expressing neuronal population in the dorsomedial hypothalamus/dorsal hypothalamic area (DMH/DHA) is stimulated by ambient cold and leptin, and ablating *Lepr* in this area caused obesity in mice due to reduced energy expenditure without affecting food intake (*Zhang et al., 2011*; *Rezai-Zadeh et al., 2014*), revealing critical function of leptin signaling in the DMH/DHA on energy expenditure. Because both POA and DMH/DHA Lepr neurons control BAT thermogenesis to modulate energy expenditure, it is plausible that thermoregulation-related Lepr neurons integrate information on ambient temperature and internal energy state that is represented by leptin to fine tune the energy expenditure level. Taken together, we hypothesized that POA leptin signaling contributes to energy balance and body weight homeostasis by modulating energy expenditure and/or food intake in interaction with ambient temperature information. This possibility is further supported by studies showing that leptin is required for ambient temperature-dependent food intake adjustment (*Kaiyala et al., 2015*; *Kaiyala et al., 2016*).

In the current study, we investigated the role of POA leptin signaling by selective stimulation or knockdown of POA *Lepr* in mice. We demonstrate that POA leptin signaling contributes to body weight homeostasis by modulating energy expenditure level, specifically under a surplus energy state, in which circulating leptin level is high (*Münzberg et al., 2004*). However, neither stimulation nor blockage of POA leptin signaling affected acute temperature-dependent metabolic adaptations. Finally, our study implicates Lepr$^{POA}$ neurons in the HFD-induced increase of metabolic rate.

## Results

### Lepr$^{POA}$ neurons respond heterogeneously to leptin

First, we directly measured the effect of leptin on membrane potential of Lepr$^{POA}$ neurons using whole-cell patch clamp recordings in *Lepr$^{EGFP}$* reporter mice. In the absence of synaptic transmission blockers, a puff application of leptin depolarized most Lepr$^{POA}$ neurons (56%), while less neurons were hyperpolarized (22%) or did not change the membrane potential (22%)(*Figure 1A,C*). We obtained a similar result using a bath application of leptin (*Figure 1—figure supplement 1*). Leptin treatment in the presence of synaptic transmission blockers (1 µM TTX +30 µM AP5 +30 µM CNQX +50 µM bicuculline) reduced the portion of depolarized neurons by 20%, indicating possible inputs from presynaptic Lepr neurons onto Lepr$^{POA}$ neurons (*Figure 1B,D*). The percentage of hyperpolarized neurons remained unaffected. These results show that Lepr$^{POA}$ neurons represent a heterogeneous population that responds differentially to leptin.

### POA leptin signaling transiently suppresses food intake

We tested next how stimulating POA Lepr affects food intake or energy expenditure. To this end, we chronically implanted a guide cannula in the POA to enable local delivery of leptin (*Figure 2A*, *Figure 2—figure supplement 1A*). First, we tested the effect of intra-POA leptin (0.05 or 0.5 pg) (*Ishihara et al., 2004*; *O'Doherty and Nguyen, 2004*; *Page-Wilson et al., 2013*) on normal dark cycle feeding (injections done 1 hr before the dark cycle) and measured food intake overnight (*Figure 2B*). Leptin significantly suppressed food intake in a dose-dependent manner during the first 3 hr after injection (*Figure 2C*). However, this leptin effect was transient and did not significantly affect total overnight food intake (*Figure 2D–E*). We confirmed that this leptin effect was not due to the leptin leakage to other areas, especially the arcuate nucleus of the hypothalamus (ARC), by staining for the phosphorylation of signal-transducer-and-activator-of-transcription-3 (pSTAT3), a marker for functional Lepr signaling (*Figure 2—figure supplement 1B*).

In the same mice, we investigated the role of POA leptin signaling (0.5 pg) during temperature-dependent metabolic adaptations using indirect calorimetry. Energy expenditure was compared between treatments at three different ambient temperatures (22, 30, and 10°C) during the light phase. While acute cold and warm exposure caused expected increases and decreases in energy expenditure, respectively, intra-POA leptin did not affect energy expenditure at any temperature condition tested (*Figure 3A–D*). Similarly, food intake increased during 6 hr of cold exposure and decreased during 6 hr of warm exposure but intra-POA leptin injection had no significant effect on

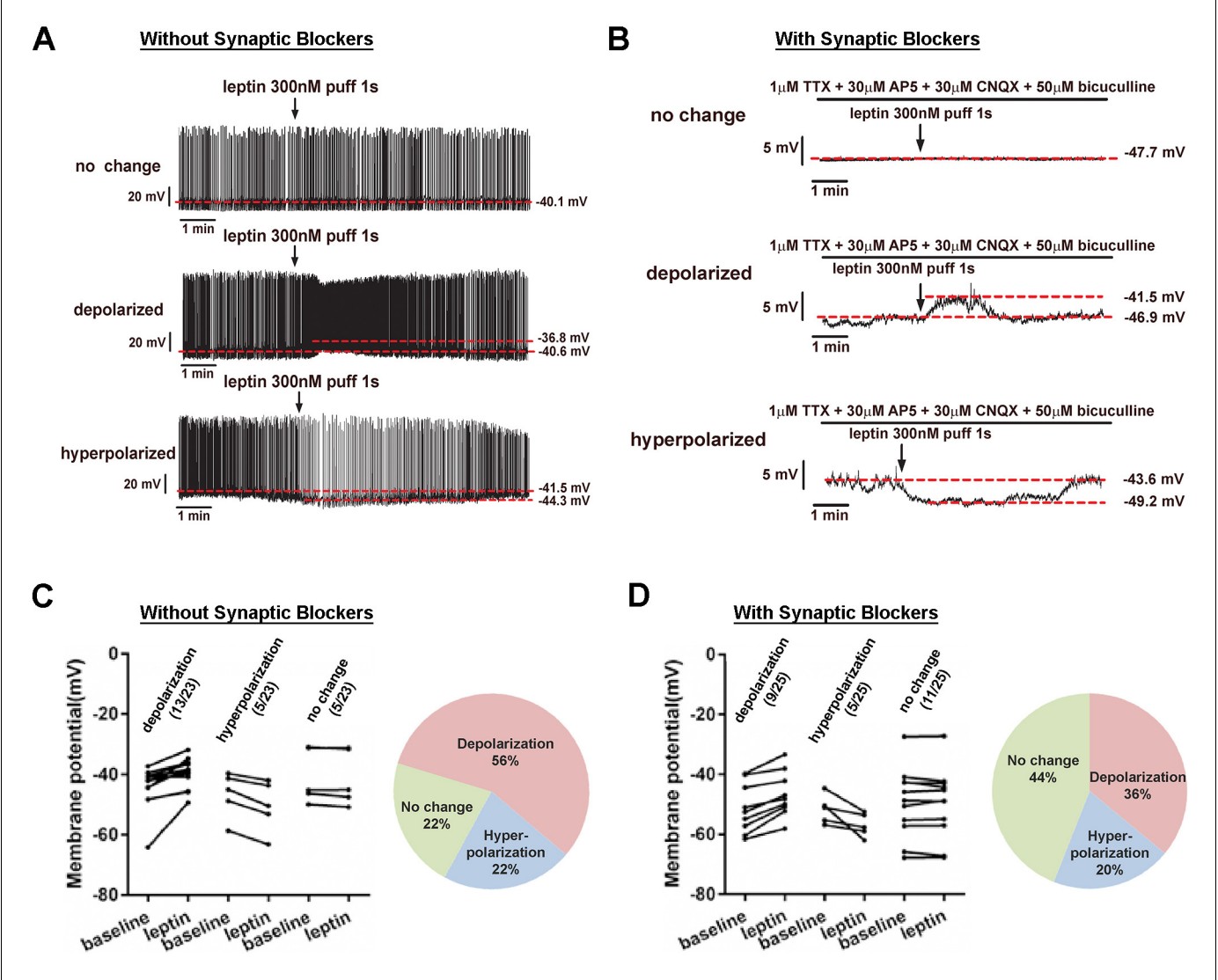

**Figure 1.** Lepr[POA] neurons respond heterogeneously to leptin. (A–B) Representative whole-cell current clamp recordings of Lepr[POA] neurons showing three different responses to puff application of leptin in the absence (A) and the presence (B) of synaptic transmission blockers. (C–D) Membrane potential changes of individual Lepr[POA] neurons in response to puff application of leptin in the absence (C, n = 23) and the presence (D, n = 25) of synaptic transmission blockers and the pie charts showing percetages of depolarization, hyperpolarization, and no change.

DOI: https://doi.org/10.7554/eLife.33505.002

The following source data and figure supplement are available for figure 1:

**Source data 1.** Membrane potentials of individually recorded Lepr[GFP] neurons at baseline and following leptin treatment used to categorize leptin response profiles in *Figure 1C,D* and *Figure 1—figure supplement 1A*.

DOI: https://doi.org/10.7554/eLife.33505.004

**Figure supplement 1.** Whole-cell current clamp of Lepr[POA] neurons in response to bath application of leptin.

DOI: https://doi.org/10.7554/eLife.33505.003

temperature-dependent food intake adjustment (*Figure 3E*). These data collectively indicate that POA leptin signaling suppresses dark-onset food intake but is not involved in acute metabolic adaptations to ambient temperature changes.

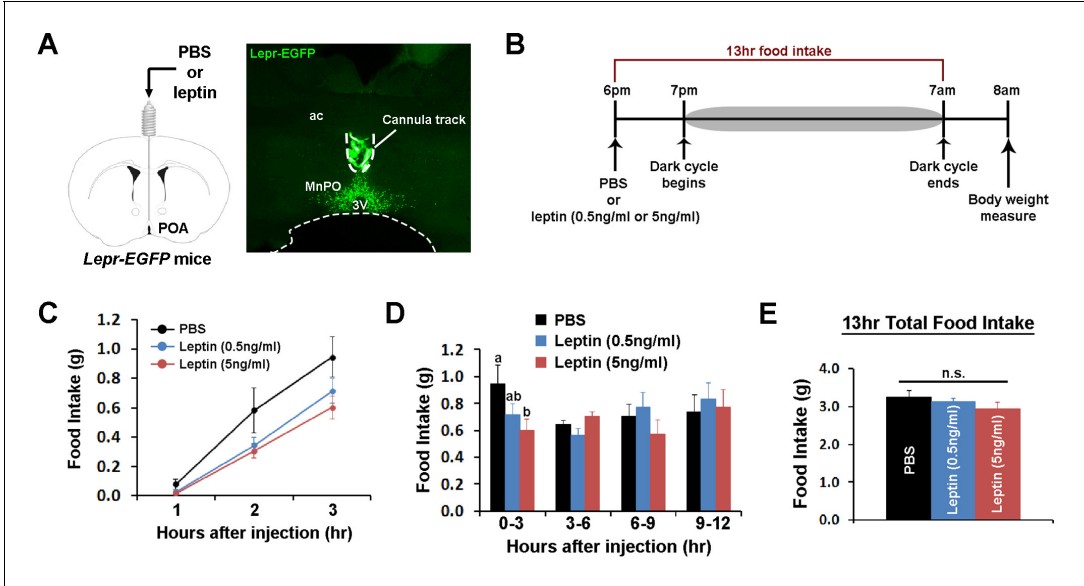

**Figure 2.** POA leptin signaling suppresses food intake. (**A**) Schematic diagram showing the implantation of a chronic cannula in the POA of $Lepr^{EGFP}$ mice and a representative histological image showing the placement of the cannula. (**B**) Experimental scheme of testing POA leptin effect on food intake with BioDAQ. (**C**) Leptin injection into the POA suppressed dark-onset food intake for the first 3 hr after the injection (n = 8). There was a significant treatment effect during this period (repeated measures ANOVA, p=0.001). (**D**) POA leptin effect on food intake lasted only for the first 3 hr after the injection (n = 8). Bars with different letters denote statistical significance at p<0.05 (repeated measures ANOVA, p=0.017). (**E**) Overnight food intake was similar between treatments (n = 8; repeated measures ANOVA, p=0.374). 3V, third ventricle; ac, anterior commissure; MnPO, median preoptic nucleus.

DOI: https://doi.org/10.7554/eLife.33505.005

The following source data and figure supplement are available for figure 2:

**Source data 1.** Raw hourly food intake data over 13 hr and cumulative food intake data for time bins following intra-POA injections of PBS and leptin used to generate *Figure 2C,D,E*.

DOI: https://doi.org/10.7554/eLife.33505.007

**Figure supplement 1.** Histological analysis of injection sites and pSTAT3 specificity in the hypothalamus.

DOI: https://doi.org/10.7554/eLife.33505.006

## POA leptin signaling is necessary for fasting-induced metabolic adaptations

We then studied cell autonomous leptin function in Lepr$^{POA}$ neurons by selective knockdown of *Lepr* in the POA (*Lepr$^{POA}$* KD). We injected Cre-expressing adeno-associated virus (AAV-GFP:Cre) or green fluorescent protein-expressing control AAV (AAV-GFP) in the POA of *Lepr$^{flox/flox}$* mice (*Figure 4A*)(*McMinn et al., 2004*). Retrospective histological analysis of pSTAT3 revealed that we obtained about 80% *Lepr* KD in the POA (*Figure 4B*, *Figure 4—figure supplement 1*).

Mice were fed regular chow diet (13.5 kcal% fat) and body weight and food intake were measured weekly for 3 weeks before viral injection and 10 weeks post-injection. Throughout the study, body weight and food intake did not differ between groups (*Figure 4C,D*). Energy expenditure was measured once before and three times after the viral injection but did not show any group difference at any stage (*Figure 4E*). Temperature-dependent changes in energy expenditure and food intake at cold (10°C) or warm (30°C) ambient temperature were also unaffected by POA *Lepr* KD, similar to POA Lepr stimulation (data not shown). We also did not see sex differences in any measurement (*Figure 4—figure supplement 2A–F*). Together, the results suggest that POA Lepr is not required for body weight homeostasis or ambient temperature-dependent metabolic adaptations.

However, based on leptin's well-known ability to inhibit fasting-induced hypometabolism in mice (*Figure 4F*)(*Döring et al., 1998*; *Geiser et al., 1998*) and weight-reduced humans (*Rosenbaum and Leibel, 2010*), we further tested whether POA Lepr is involved in exogenous leptin effects on energy expenditure during a state of negative energy balance. Leptin was unable to prevent fasting-induced hypometabolism in *Lepr$^{POA}$* KD mice contrary to control mice. This was primarily due to a failure to

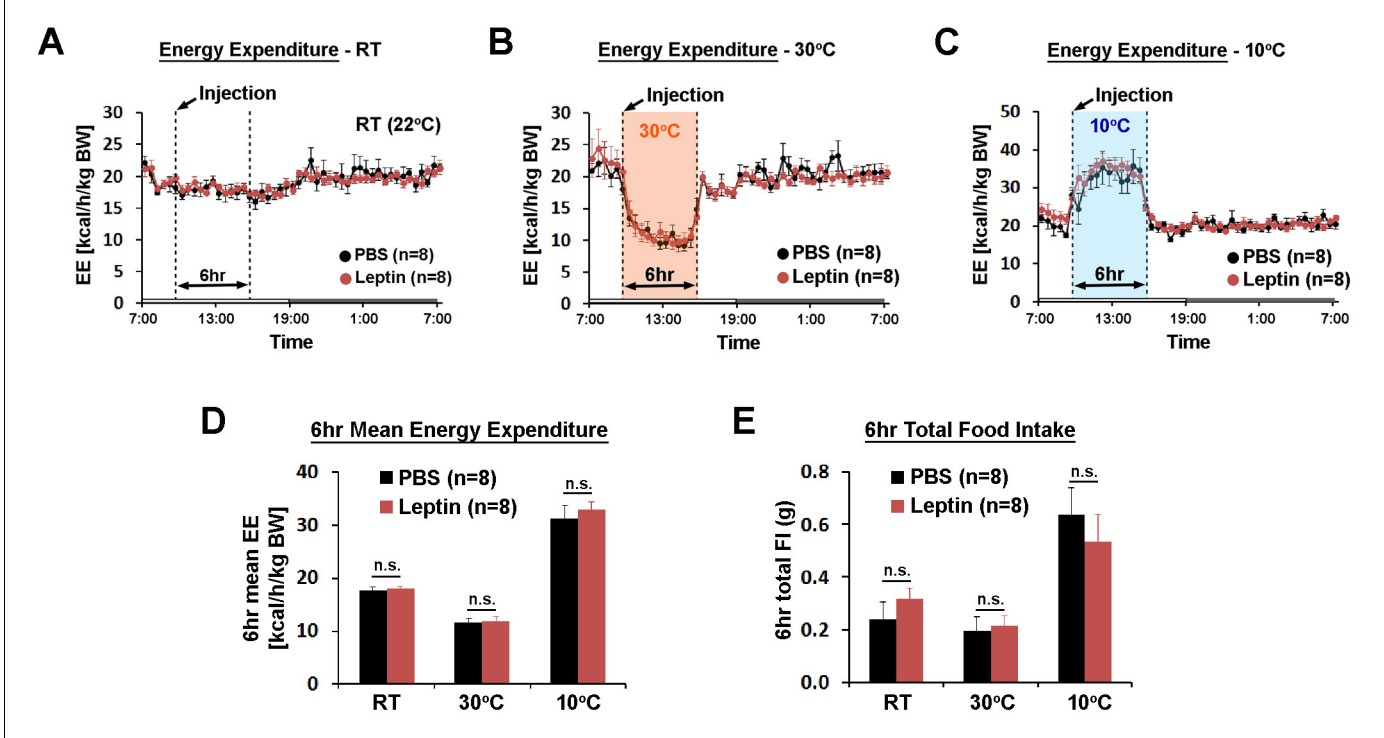

**Figure 3.** POA leptin signaling does not affect ambient temperature-dependent metabolic adaptations. (A–C) 24 hr energy expenditure from PBS and leptin injections (5 ng/ml, total 0.5 pg) are overlaid for RT (22°C), warm (30°C), and cold (10°C) (n = 8 each). Temperature was changed for 6 hr from 9 am to three pm. (D) 6 hr mean energy expenditure was compared between treatment conditions (paired *t*-test between PBS and leptin at each temperature). (E) Total food intake during the same 6 hr as in (D).

DOI: https://doi.org/10.7554/eLife.33505.008

The following source data is available for figure 3:

**Source data 1.** Raw and mean energy expenditure for generation of *Figure 3A,B,C and D*; and food intake data used to generate *Figure 3E*.
DOI: https://doi.org/10.7554/eLife.33505.009

properly lower energy expenditure in response to a negative energy state during fasting (*Figure 4G*). As a result, there was no leptin-induced recovery of energy expenditure in *Lepr^POA* KD mice.

## POA leptin signaling is involved in body weight regulation under high fat diet

The above data suggested that POA Lepr may be important for adjusting metabolic rate in response to internal energy state. To test this possibility, we used HFD because this diet results in a positive energy balance that leads to body weight gain despite increased energy expenditure, the opposite outcome of fasting (*Figure 4F*). In a new cohort of mice, we repeated the same experiment but the diet was switched from chow to HFD (58 kcal% fat) following viral injections. At the end of this study, we verified about 50% *Lepr* KD based on pSTAT3 immunohistochemical analysis (*Figure 5—figure supplement 1*).

Mice injected with AAV-GFP:Cre in the POA gained about twice as much weight as mice injected with AAV-GFP by 15 weeks after viral injection (*Figure 5A*). This increase in body weight was mainly due to increased fat mass, as shown by a trend for increased adiposity (*Figure 5B*, *Figure 5—figure supplement 2A,B*). The level of pSTAT3 in the POA varied significantly in mice injected with AAV-GFP:Cre due to inherent variability of viral injections, which may explain the large standard errors for this group in *Figure 5A–B*. Thus, we correlated the body weight change at week 15 and the adiposity index at week 13 to POA pSTAT3+ cell numbers across all mice, and both parameters showed significant negative correlation with POA pSTAT3+ cell numbers (*Figure 5C,D*). These analyses imply

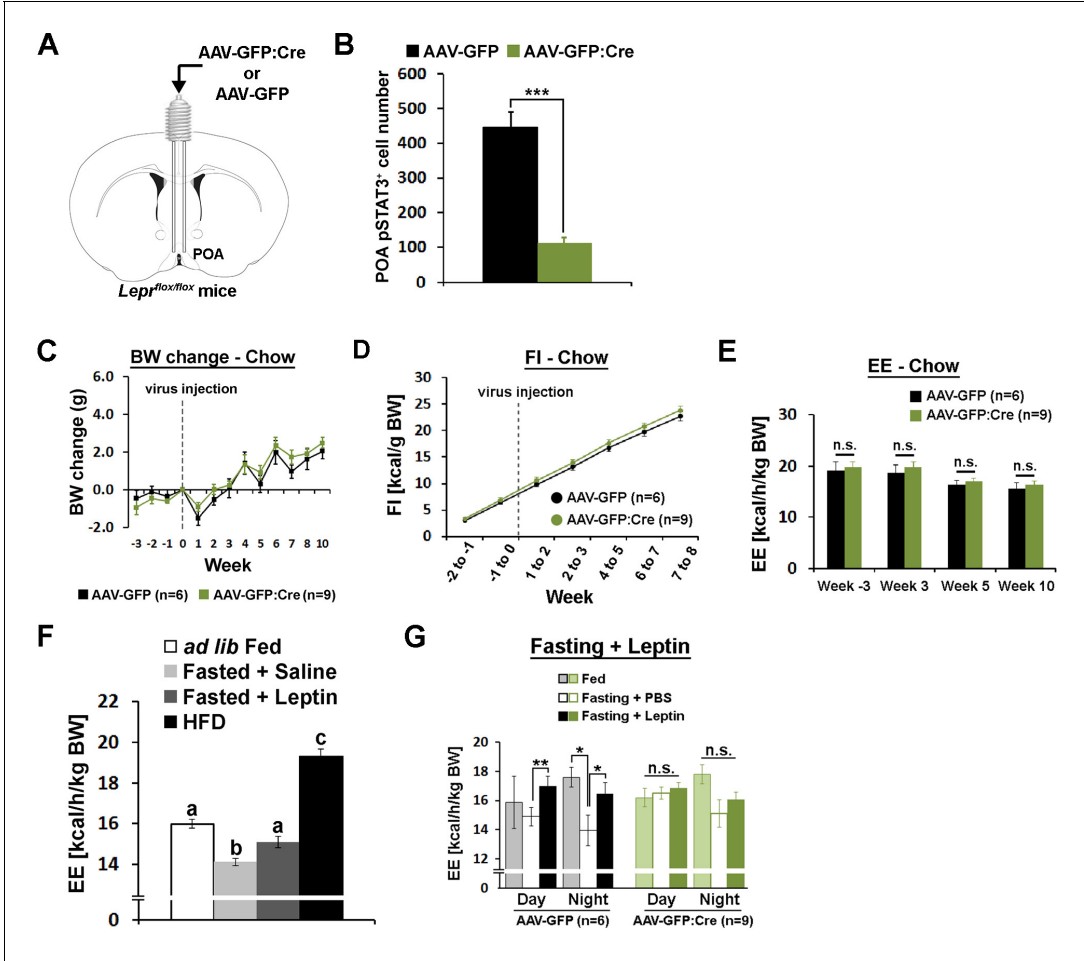

**Figure 4.** POA Lepr is necessary for fasting-induced hypometabolism. (**A**) A schematic diagram showing bilateral viral injection into the POA of *Lepr*$^{flox/flox}$ mice. (**B**) AAV-GFP:Cre injection resulted in about 80% reduction of pSTAT3$^{+}$ cell numbers in the POA (n = 6 for AAV-GFP, n = 9 for AAV-GFP:Cre; independent *t*-test, p<0.001). (**C–E**) Body weight, food intake, and energy expenditure were similar between groups throughout the study (repeated measures ANOVA followed by Bonferroni pairwise comparisons, p>0.05). (**F**) Energy expenditure comparison between *ad lib* fed (n = 10), fasted +saline (n = 5), fasted +leptin (n = 5; 5 mg/kg, i.p.), and 2-week-long HFD feeding (n = 5). Leptin injection attenuated fasting-induced hypometabolism while HFD increased the metabolic rate. Bars with different letters denote statistical significance at p<0.05 (one-way ANOVA followed by Bonferroni pairwise comparisons). (**G**) In *Lepr*$^{POA}$ KD mice, fasting did not decrease energy expenditure, thus leptin failed to attenuate fasting-induced hypometabolism (repeated measures ANOVA followed by Bonferroni pairwise comparisons, *p<0.05, **p<0.01).

DOI: https://doi.org/10.7554/eLife.33505.010

The following source data and figure supplements are available for figure 4:

**Source data 1.** Individual cell counts used to generate *Figure 4B*; raw body weight and body weight changes used to generate *Figure 4C*; raw food intake and weight specific caloric intake used to generate *Figure 4D*; raw and mean energy expenditure used to generate *Figure 4E,F,G*.
DOI: https://doi.org/10.7554/eLife.33505.013

**Figure supplement 1.** Histological analysis of virus spread and pSTAT3.
DOI: https://doi.org/10.7554/eLife.33505.011

**Figure supplement 2.** Body weight, food intake, and energy expenditure by sex.
DOI: https://doi.org/10.7554/eLife.33505.012

that POA Lepr signaling is important for body weight control under HFD conditions. The increase in body fat and weight was not due to increased food consumption because mice from both groups ate the same amount of food throughout the study (*Figure 5E*, *Figure 5—figure supplement 2C*).

Energy expenditure, measured once before and twice after viral injection, showed no significant group difference at any time point (*Figure 5—figure supplement 2D*). However, there was a trending decrease of energy expenditure in *Lepr*$^{POA}$ KD mice from week −3 to week 10 (*Figure 5F*), which resulted in a bigger energy expenditure reduction from week −3 to week 10, compared to

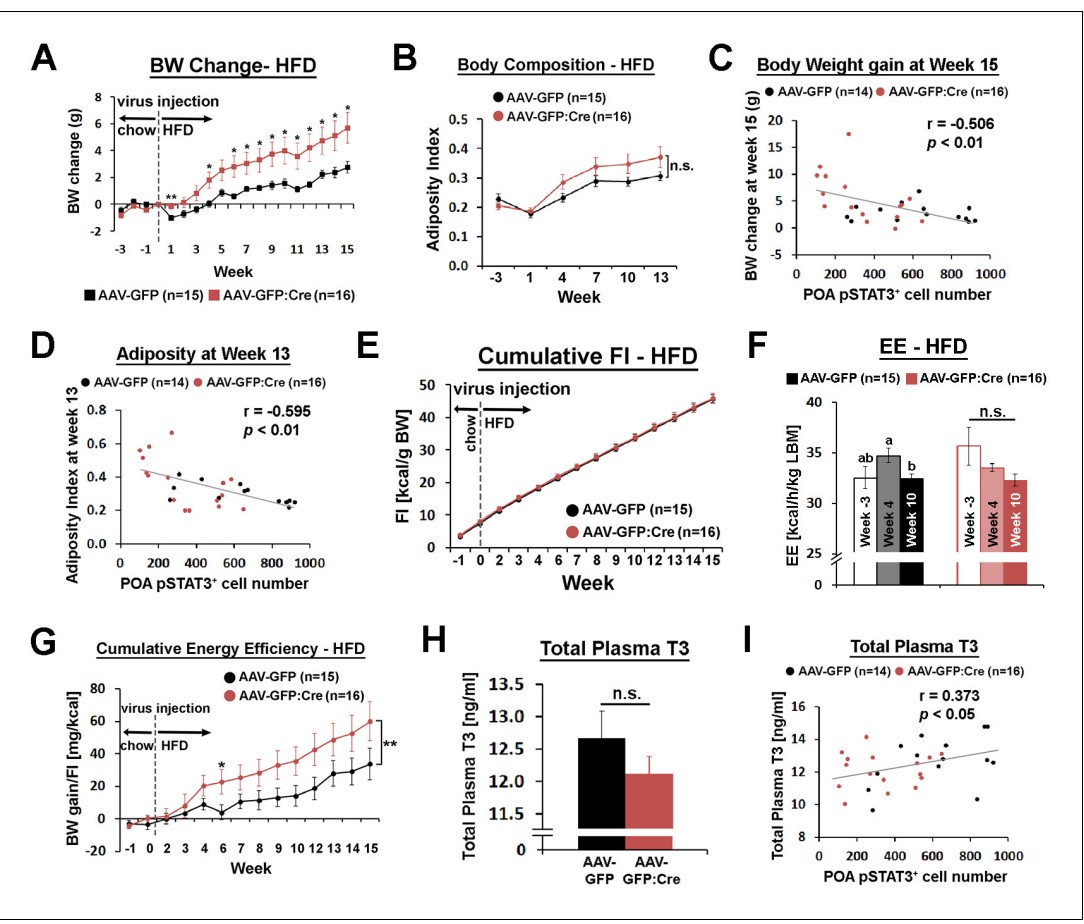

**Figure 5.** POA Lepr is involved in body weight regulation under HFD. (**A**) *Lepr^POA* KD mice gained greater body weight than control mice over 15 weeks on HFD (repeated measures ANOVA followed by Bonferroni pairwise comparisons, *p<0.05, **p<0.01). (**B**) There was a trend of higher adiposity index (fat mass/lean mass) in *Lepr^POA* KO mice (the effect of virus type on adiposity index was tested by repeated measures ANOVA, p=0.141). (**C–D**) Body weigh change at week 15 and adiposity index at week 13 are negatively correlated with POA pSTAT3[+] cell numbers (Pearson correlation). (**E**) Cumulative food intake during the entire study was similar between groups (repeated measures ANOVA followed by Bonferroni pairwise comparisons). (**F**) Energy expenditure is compared between time points within each group. Bars with different letters denote statistical significance at p<0.05 (repeated measures ANOVA followed by Bonferroni pairwise comparisons). (**G**) Cumulative energy efficiency (body weight gain/food intake) showed a significant interaction between week and virus type ($F_{(11,319)}$ = 2.518, p<0.01, repeated measures ANOVA followed by Bonferroni pairwise comparisons, *p<0.05). (**H**) Total plasma T3 levels were not significantly different between groups (independent *t*-test, p=0.267). (**I**) Total plasma T3 levels showed a significant positive correlation with POA pSTAT3[+] cell numbers (Pearson correlation).

DOI: https://doi.org/10.7554/eLife.33505.014

The following source data and figure supplements are available for figure 5:

**Source data 1.** Raw data body weight, raw body composition, body weight changes, adiposity index and percent body composition used to generate *Figure 5A,G* and *Figure 5—figure supplement 2A and B*; food intake used to generate *Figure 5E andG*; raw and mean energy expenditure used to generate *Figure 5F* and *Figure 5—figure supplement 2E*; energy efficiency used to generate *Figure 5G*; raw scoring and total number of estrous cycles used to generate *Figure 5—figure supplement 2I*.

DOI: https://doi.org/10.7554/eLife.33505.019

**Figure supplement 1.** Histological analysis of virus spread and pSTAT3.

DOI: https://doi.org/10.7554/eLife.33505.015

**Figure supplement 2.** POA Lepr is involved in body weight regulation under HFD.

DOI: https://doi.org/10.7554/eLife.33505.016

**Figure supplement 3.** BAT histological analysis and POA leptin effect on total plasma T3.

DOI: https://doi.org/10.7554/eLife.33505.017

*Figure 5 continued on next page*

*Figure 5 continued*

**Figure supplement 4.** Projection of Lepr^POA neurons to the PVH.
DOI: https://doi.org/10.7554/eLife.33505.018

control mice (*Figure 5—figure supplement 2E*). Still, these comparisons were not statistically significant, and the baseline energy expenditure at week 10 and POA pSTAT3$^+$ cell number did not show a significant correlation (*Figure 5—figure supplement 2F*). A main caveat of the cross-sectional measurement of energy expenditure is that it is prone to miss a very small difference over a long period of time. Therefore, we used a more longitudinal measure that can reflect changes in energy expenditure, that is an energy efficiency expressed as body weight gain divided by food intake, cumulated throughout the entire study (*Figure 5G*). Energy efficiency also showed a weak but significant negative correlation with POA pSTAT3$^+$ cell number (*Figure 5—figure supplement 2G*), suggesting that *Lepr^POA* KD mice spent less energy per unit food consumed.

Because Lepr^POA neurons control BAT SNA, we first tested whether BAT function declined in *Lepr^POA* KD mice. Expression of BAT function-related genes, *uncoupling protein 1 (Ucp1)*, *peroxisome proliferative activated receptor gamma, coactivator one alpha (Ppargc1a, also known as Pgc1a)*, and *deiodinase 2 (Dio2)* showed a trending decrease in *Lepr^POA* KD mice (*Figure 5—figure supplement 2H*), and expression levels of these genes were not significantly correlated with the POA pSTAT3$^+$ cell number (data not shown). Consistent with this result, there was slightly increased but statistically non-significant increase in fat accumulation in BAT in *Lepr^POA* KD mice (*Figure 5—figure supplement 3A*). We next examined if the thyroid axis was affected in *Lepr^POA* KD mice because systemic leptin increases *thyrotropin releasing hormone (Trh)* gene expression (*Nillni et al., 2000*; *Harris et al., 2001*; *Guo et al., 2004*) and the POA has been implicated in TRH secretion (*Andersson et al., 1963*; *Martelli et al., 2014*). TRH secretion ultimately leads to thyroid hormone secretion and metabolic rate increase through both central and peripheral mechanisms (*López et al., 2013*; *Mullur et al., 2014*). We found that plasma levels of total triiodothyronine (T3) was only slightly and not significantly lower in *Lepr^POA* KD mice but significantly correlated with the POA pSTAT3$^+$ cell number (*Figure 5H,I*), suggesting that POA Lepr stimulates the thyroid axis to modulate energy expenditure. To corroborate this result, we observed strong projection of Lepr^POA neurons to the paraventricular nucleus of the hypothalamus (PVH) in the field of TRH neurons (*Figure 5—figure supplement 4A–B*). Furthermore, when wheat germ agglutinin (WGA)-GFP, an anterograde transsynaptic tracer, was expressed in Lepr^POA neurons, we observed a rare case of its expression in a TRH neuron in the PVH (*Figure 5—figure supplement 4C–D*). However, intra-POA leptin injection (0.5 pg) in fasted mice failed to significantly elevate the total T3 level (*Figure 5—figure supplement 3B*), unlike the effect of systemic exogenous leptin in fasted mice on the thyroid hormone level (*Ahima et al., 1996*; *Ahima et al., 1999*). Consistent with POA *Lepr* KD under chow diet, ambient temperature-dependent metabolic adaptations in HFD-fed *Lepr^POA* KD mice were not affected (data now shown).

Finally, POA Lepr is implicated in reproductive function in female mice by promoting nitric oxide synthesis that results in luteinizing hormone (LH) release (*Bellefontaine et al., 2014*). We tested the effect of POA *Lepr* KD on female reproduction by measuring the estrous cycles for 19 days. The total number of cycles during this period was similar between groups (*Figure 5—figure supplement 2I*).

## Discussion

Our study demonstrates that POA Lepr is involved in body weight regulation by modulating energy expenditure under negative (fasting) or positive (HFD) energy balance without affecting food intake, which may involve the control of BAT thermogenesis and thyroid hormone release. In contrast, a direct leptin injection into the POA transiently suppresses food intake without affecting energy expenditure. There are at least a couple of possible explanations for the seemingly contradictory phenotypes between the POA leptin injection and *Lepr* deletion. The first possibility is the well-known discrepancy of leptin function between below and above the normal physiological leptin level. The lack of leptin is generally considered a stronger signal than increasing leptin levels above normal circulating levels and biological responses do not show a linear relationship with the leptin

concentration (*Leibel, 2002*). Another possibility is leptin's action on different cell types. In other words, the effect on food intake may be mediated by presynaptic Lepr expressed in neurons innervating the POA, likely at axon terminals, while the effect on energy expenditure may be mediated cell autonomously in POA Lepr neurons. A similar dual mode of leptin action exists in the ARC between agouti related neuropeptide (AGRP) and pro-opiomelanocortin (POMC) neurons, in which leptin suppresses the presynaptic inhibitory AGRP neuronal input onto POMC neurons and cell autonomously stimulates POMC neurons (*Cowley et al., 2001*). Our electrophysiological recordings also support this notion because leptin application in the presence of a cocktail of synaptic transmission blockers reduced the percentage of depolarized neurons compared to leptin application in the absence of blockers. The POA receives inputs from many Lepr-expressing neurons that could modulate food intake including AGRP and POMC neurons (*Wang et al., 2015*).

Our POA *Lepr* KD study suggests that leptin acts on Lepr$^{POA}$ neurons to modulate energy expenditure. However, because the energy expenditure phenotype of POA *Lepr* KD was only evident under fasting or HFD feeding and the effect was small, an acute intra-POA leptin injection in *ad lib* chow-fed mice may have failed to affect energy expenditure or induced undetectable changes. A more chronic leptin administration into the POA in food-restricted mice or *ob/ob* mice may be required in order to detect this small and specific leptin effect on energy expenditure. Furthermore, the retrospective histological pSTAT3 analysis showed that only a portion of POA Lepr had been stimulated by intra-POA leptin injection.

Because Lepr$^{POA}$ neurons control ambient temperature-dependent metabolic adaptations (*Yu et al., 2016*), we first speculated that POA leptin function may be involved in this aspect. This reasoning was further supported by the fact that leptin gene expression is inversely proportional to the SNA level to adipose tissue (*Trayhurn et al., 1995*; *Evans et al., 1999*), which is controlled by ambient temperature and the POA. However, neither stimulation nor ablation of POA *Lepr* affected temperature-dependent adaptive changes in energy expenditure and food intake. Instead, energy expenditure was specifically affected when the internal energy state, which is reflected by circulating leptin levels, was low or high in *Lepr$^{POA}$* KD mice. Although it seems counterintuitive that *Lepr$^{POA}$* KD mice have higher energy expenditure during fasting than control mice, a similar discordance of leptin-linked physiology was reported for female reproduction. In female *Lepr$^{flox/flox}$* mice injected with AAV-Cre in the POA, the baseline LH level was higher than that of control mice and leptin failed to further induce LH secretion (*Bellefontaine et al., 2014*), reminiscent of our energy expenditure phenotype in *Lepr$^{POA}$* KD mice during fasting. Despite the increased LH level in POA *Lepr*-ablated mice, female reproduction did not seem to be affected based on our study.

The exact cellular mechanism of how leptin affects Lepr$^{POA}$ neuron functions is not clear due to small effect sizes and trending but statistically non-significant results for various measurements for energy expenditure. While Lepr$^{POA}$ neurons strongly modulate BAT thermogenesis (*Yu et al., 2016*), our results show no significant changes in BAT function in *Lepr$^{POA}$* KD mice. However, data from BAT gene expression and histology all point toward small but blunted BAT function by *Lepr$^{POA}$* KD. The weak but significant positive correlation between POA STAT3 activation and plasma T3 is consistent with previous studies showing leptin's involvement in expression and secretion of TRH (*Andersson et al., 1963*; *Nillni et al., 2000*; *Martelli et al., 2014*) and ability to prevent the fasting-induced drop in the thyroxine level (*Ahima et al., 1996*). We have also observed strong projections of Lepr$^{POA}$ neurons to the PVH in the field of TRH neurons, even though we found little evidence for direct synaptic connection from Lepr$^{POA}$ neurons to TRH$^{PVH}$ neurons. Nonetheless, these collective data do not prove that POA Lepr signaling modulates the thyroid axis. It requires further investigation if neuronal (thermosensory) and humoral (leptin) inputs to Lepr$^{POA}$ neurons modulate energy expenditure through separate mechanisms.

Lastly, it is important to point out that our AAV-GFP:Cre injection into the POA achieved only about 50% *Lepr* KD on average in the HFD study. We believe this partial ablation of POA Lepr and inherent variability of viral injections contributed to relatively small effect sizes and big standard errors that rendered many measurement differences between groups statistically insignificant. Therefore, the contribution of POA leptin signaling to energy balance and body weight homeostasis may be underestimated in our study.

Our study is the first to link POA leptin signaling to body weight control. HFD promotes a higher metabolic rate and POA Lepr may be necessary for this HFD-induced increase in energy expenditure. Therefore, the loss of POA Lepr may have accelerated the body weight gain caused by HFD in

*Lepr^POA* KD mice. Our study also proposes a potential integration of environmental temperature information and internal energy state in the POA through neural and humoral routes, respectively. Even though acute temperature-dependent metabolic adaptations did not interact with POA leptin signaling, the long-term interaction still remains to be investigated.

# Materials and methods

**Key resources table**

| Reagent type (species) or resource | Designation | Source or reference | Identifiers | Additional information |
|---|---|---|---|---|
| strain, strain background (*M. Musculus*) | C57BL/6J | The Jackson Laboratory | JAX: 000664; RRID: IMSR_JAX:000664 | |
| genetic reagent (*M. Musculus*) | *Lepr^EGFP* | Dr. Martin G. Myers, Jr., University of Michigan; PMID: 17021368 | | Full nomenclature: *Lepr^Cre/Cre*; *Gt(ROSA)26Sor^tm2Sho/tm2Sho* |
| genetic reagent (*M. Musculus*) | *Lepr^flox/flox* | Dr. Streamson C. Chua, Jr., Albert Einstein College of Medicine; PMID: 15389315 | | |
| genetic reagent (*Adeno-associated virus*) | AAV-GFP (AAV5-hSyn-EYFP) | University of North Carolina Vector Core | UNC: AV4836 | Titer: 4.0 × 10Exp12 vg/ml |
| genetic reagent (*Adeno-associated virus*) | AAV-GFP:Cre (AAV5-hSyn-GFP:Cre) | University of North Carolina Vector Core | UNC: AV6446 | Titer: 5.1 × 10Exp12 vg/ml |
| antibody | anti-pSTAT3 (Tyr705, rabbit polyclonal) | Cell Signaling | Cell Signaling: 9131RRID: AB_331586 | (1:500) |
| antibody | anti-GFP (chicken polyclonal) | Abcam | Abcam: ab13970; RRID: AB_300798 | (1:1000) |
| antibody | anti-TRH (rabbit polyclonal) | Dr. Eduardo A. Nillni, Brown University | EAN: pYE26 | (1:1000) |
| antibody | anti-UCP1 | Abcam | Abcam: ab209483 RRID: AB_2722676 | (1:4000) |
| antibody | anti-rabbit IgG-biotin (donkey polyclonal) | ImmunoResearch Laboratories | IRL: 711-065-152; RRID: AB_2340593 | (1:1000) |
| antibody | anti-chicken IgY-DyLight 488 (donkey polyclonal) | ImmunoResearch Laboratories | IRL: 703-486-155 discontinued | (1:200) |
| antibody | anti-rabbit AlexaFluor 594 (donkey polyclonal) | ThermoFisher Scientific | TFS: A-21207; RRID: AB_141637 | (1:200) |
| peptide, recombinant protein | leptin (mouse recombinant) | National Hormone and Peptide Program | | http://www.humc.edu/hormones |
| commercial assay or kit | Milliplex Map Kit (plasma T3) | Millipore Sigma | Millipore Sigma: RTHYMAG-30K | |
| commercial assay or kit | Vectastain ABC kit | Vector Laboratories | Vecotr Laboratories: PK-6100 | |
| other | HFD (high fat diet for mouse) | Research Diets | Research Diets: D12331 | 58 kcal% fat |

## Mice

All experiments were approved by the Institutional Animal Care and Use Committee at Pennington Biomedical Research Center, Baylor College of Medicine, and Tulane University. Mice were housed at 22–24°C with a 12:12 light:dark cycle. Laboratory rodent diet (5001, LabDiet) and water were available *ad libitum* unless otherwise stated. *Lepr^EGFP* and *Lepr^Cre* mice were kindly provided by Dr. Martin G. Myers, Jr., University of Michigan (*Leshan et al., 2010*; *Leshan et al., 2006*) and *Lepr^flox/flox* mice by Dr. Streamson C. Chua, Jr., Albert Einstein College of Medicine (*McMinn et al., 2004*). C57BL/6J mice were obtained from the Jackson Laboratory (stock #000664).

## Electrophysiology

*Lepr^EGFP* mice were used for all neuronal activity recordings. Mice were deeply anesthetized with iso-flurane and transcardially perfused with a modified ice-cold artificial cerebral spinal fluid (aCSF: 10

mM NaCl, 25 mM NaHCO$_3$, 195 mM Sucrose, 5 mM Glucose, 2.5 mM KCl, 1.25 mM NaH$_2$PO$_4$, 2 mM Na-pyruvate, 0.5 mM CaCl$_2$, 7 mM MgCl$_2$) (*Cao et al., 2014*). The mice were then decapitated, and the entire brain was removed. Brains was quickly sectioned in ice-cold aCSF solution (126 mM NaCl, 2.5 mM KCl, 1.2 mM MgCl$_2$, 2.4 mM CaCl$_2$, 1 mM NaH$_2$PO$_4$, 5.0 mM Glucose, and 21.4 mM NaHCO$_3$) saturated with 95% O$_2$ and 5% CO$_2$. Coronal sections containing the POA (250 µm) was cut with a Microm HM 650V vibratome (Thermo Scientific). Then the slices were recovered in the aCSF at 34°C for 1 hr. Whole-cell patch clamp recordings were performed in the GFP-labelled mature POA neurons visually identified by an upright microscope (Eclipse FN-1, Nikon) equipped with IR-DIC optics (Nikon 40x NIR). Signals were processed using Multiclamp 700B amplifier (Axon Instruments), sampled using Digidata 1440A and analyzed offline on a PC with pCLAMP 10.3 (Axon Instruments). The slices were bathed in oxygenated aCSF (32–34°C) at a flow rate of approximately 2 ml/min. Patch pipettes with resistances of 3–5 MΩ were filled with solution containing 126 mM K-gluconate, 10 mM NaCl, 10 mM EGTA, 1 mM MgCl$_2$, 2 mM Na-ATP and 0.1 mM Mg-GTP (adjusted to pH7.3 with KOH). Current clamp was engaged to test neural firing frequency and resting membrane potential (RM) at the baseline and after puff application of leptin (300 nM, 1 s). In some experiments, the aCSF solution also contained 1 µM tetrodotoxin (TTX) and a cocktail of fast synaptic inhibitors, namely bicuculline (50 µM, a GABA receptor antagonist), DAP-5 (30 µM, an NMDA receptor antagonist) and CNQX (30 µM, an AMPA receptor antagonist) to block the majority of presynaptic inputs. The values for RM and firing frequency were averaged within 2 min bin at the baseline or after leptin puff. The RM values were calculated by Clampfit 10.3. A neuron was considered depolarized or hyperpolarized if a change in membrane potential was at least 2 mV in amplitude and this response was observed after leptin application and stayed stable for at least 1 min.

## Viruses and stereotaxic surgeries

Stereotaxic injection of AAV was performed as previously described (*Rezai-Zadeh et al., 2014*). Briefly, 12–14 week-old *Lepr*^flox/flox^ mice were placed on a stereotaxic alignment system (1900, David Kopf Instruments) and maintained anesthetized by 2% isoflurane during surgeries. For the selective deletion of *Lepr* in the POA, AAV5-hSyn-EYFP (AAV-GFP, 4.0 × 10$^{12}$ vg/ml) or AAV5-hSyn-GFP:Cre (AAV-GFP:Cre, 5.1 × 10$^{12}$ vg/ml) was injected into the POA (AP:+0.6 mm, ML:±0.4 mm, DV: −5.2 mm; 200 nl or 400 nl total at 20 nl/30 s) bilaterally using a bilateral guide cannula and injector set (C235G, Plastics One). Both viruses were obtained from the vector core of the University of North Carolina at Chapel Hill and injection coordinates were based on the Paxinos mouse brain atlas (*Paxinos and Franklin, 2004*).

Stereotaxic injection of Ad-iZ/EGFPf (*Leshan et al., 2009*; *Leinninger et al., 2009*) and Ad-iN/WED (*Louis et al., 2010*) in the POA was performed similarly as described above with *Lepr*^Cre^ mice. Ad-iZ/EGFPf (1.95 × 10$^{12}$ PFU/ml) expresses membrane-targeted farnesylated EGFP while Ad-iN/WED (2.0 × 10$^{12}$ PFU/ml) expresses anterograde transsynaptic WGA^GFP^ in the presence of Cre. Total 300–400 nl of viruses were injected. Brains were harvested by transcardial perfusion 10–14 days after viral injection.

For chronic cannulation in the POA, unilateral guide cannulas (C315GA, Plastics One) were implanted (AP:+0.58 mm, ML:±0 mm, DV: −4.2 mm). The guide cannulas were secured to the skull with dental cement and inserted with dummy cannulas (C315DC, Plastics One) to prevent contamination or clogs. 3-month-old male *Lepr*^EGFP^ mice (n = 8) were used for group 1 and 3-month-old male C57BL/6J mice (n = 10) were used for group 2. Mice recovered for a week before being used for experiments.

## POA leptin injection

Eight male *Lepr*^EGFP^ mice (group 1) with intra-POA cannulas were acclimated to intra-POA injections for 4 days by injecting 100 nl saline (20 nl/20 s) once per day. The injection was carried out with a 1 µl Hamilton syringe (#80100, Hamilton Company) that was connected to an injector (C315IA, Plastics One) that project 1 mm from the tip of the guide cannula. Thus, the injection depth was 5.2 mm from the bregma. During the acclimation, body weight of the mice was monitored daily and we ensured stabilization of the body weight before using mice for experiments.

We used BioDAQ (Research Diets) to test the effect of intra-POA leptin on regular chow intake. One hour before the dark cycle begins, 100 nl of PBS, 0.5 ng/ml leptin (0.05 pg), or 5 ng/ml (0.5 pg)

leptin was injected into the POA as described above and 13 hr food intake (1 hr light cycle +12 hr dark cycle) was analyzed. Leptin was obtained from Dr. Parlow (National Hormone and Peptide Program, http://www.humc.edu/hormones). See *Figure 2B* for more information on the BioDAQ experimental design. Data for PBS injection were averages of 3 replicates and data for leptin injection were averages of two replicates for each concentration. The order of injections was counterbalanced and a rest day was placed between leptin injections.

To test the effect of intra-POA leptin on energy expenditure, we used the TSE system (TSE Systems). 100 nl of PBS or 5 ng/ml leptin was injected into the POA at CT 2 hr. Injections were carried out at RT (22°C), 30 and 10°C, and data for leptin injection were averages of two injections on separate days. Injections were counterbalanced in each temperature condition. When ambient temperature was changed from RT to either 30 or 10°C, the temperature was changed from CT 2 to CT 8 hr. Energy expenditure (kcal/hr/kg BW) and food intake during those 6 hr were compared between injections and temperature conditions.

On the final experimental day of group one mice, 100 nl of 5 ng/ml leptin was injected into the POA 1.5 hr before perfusion, followed by 100 nl of red-fluorescent Fluosphere (F8801, Thermofisher scientific) to mark the injection site. Mice were deeply anesthetized with isoflurane and transcardially perfused with ice-cold saline, followed by 10% formalin. Brains were removed and post-fixed in 10% formalin overnight at 4°C and cryoprotected in 30% sucrose in PBS.

For group 2, mice were fasted for 24 hr from CT 3 hr and injected with 100 nl PBS (n = 5) or leptin (5 ng/ml; n = 5) in the POA. Injections were done three times during this period at CT 3, 11, and 3 hr next day, and brains and trunk blood were harvested 1.5 hr after the final injection.

## Selective *Lepr* KD in the POA

### Study with regular chow diet

About 3-month-old *Lepr^flox/flox* mice were injected with AAV-Cre:GFP (5 males and four females) or AAV-GFP (4 males and two females) in the POA as described above. Weekly body weight was measured from 3 weeks before viral injections to 10 weeks post-injections. Weekly food intake was measured from 2 weeks before viral injections to 8 weeks post-injections. Food intake was measured by weighing food in the food hopper at the beginning and the end of each week. Big crumbs of food pellets on the cage floor were added to hopper pellets for measuring remaining food at the end of a week. Food intake was not measured when mice were in Comprehensive Laboratory Animal Monitoring System (CLAMS; Columbus Instruments) for indirect calorimetry during week −3, week 3, week 5 and week 10 to measure energy expenditure, locomotor activity, and respiratory exchange ratio (RER). Mice were maintained on regular chow diet (13.5 kcal% fat, 5001, LabDiet) throughout the study.

During the week five calorimetry study, leptin's effect on preventing fasting-induced hypometabolism was tested by fasting mice for 24 hr from CT 5 hr and injecting them with saline or leptin (5 mg/kg, i.p.) twice during that period, at CT 5 hr and 11.5 hr. The injection order was counterbalanced.

At the end of the study, leptin (5 mg/kg, i.p) was injected 1 hr before tissue harvest for immunohistochemical analysis of leptin receptor signaling in the POA (*Laque et al., 2015*). We did not observe gender differences in any measurement, and thus presented combined data between males and females throughout the paper.

### Study with high fat diet

3-month-old *Lepr^flox/flox* mice were injected with AAV-Cre:GFP (6 males and 10 females) or AAV-GFP (6 males and nine females) in the POA. Food was switched from regular chow to high fat diet (58 kcal% fat, D12331, Research Diets) right after viral injections. Weekly body weight and food intake were measured from week −3 to week 15. Weekly food intake measurement was interrupted while mice were in TSE for indirect calorimetry at week −3, week 4 and week 10. Body composition was measured at week −3, week 1, week 4, week 7, week 10 and week 13 (Minispec LF110, Bruker). Adiposity index was calculated from fat mass divided by lean mass.

While mice were in TSE, the ambient temperature was maintained at 22°C and mice were tested for adaptive responses to ambient temperature changes. For a warm-adaptation testing, the ambient temperature was changed to 30°C from CT 11 hr to 1 hr next morning (total 14 hr). For a cold-

adaptation testing, the ambient temperature was lowered to 10°C from CT 1 hr to 9 hr (total 8 hr). Different times of the day were used for two temperature conditions to facilitate the detection of changes in energy expenditure and food intake.

We checked the estrous cycle of female mice, starting at week 5 for 19 days by checking vaginal cytology as previously described (*Caligioni, 2009*; *Byers et al., 2012*). Briefly, we flushed the vaginal cavity of female mice with sterile saline (~10 µl) 3–4 times and the final flush was collected on a glass slide. Vaginal cytology was observed unstained with a light microscope (Olympus BX51). The proestrus followed by the estrus was defined as one cycle (*Caligioni, 2009*).

On the day of tissue harvest, mice were fasted for 4 hr before leptin (5 mg/kg, i.p.) was injected 1 hr prior to tissue harvest. Brain, liver, BAT, inguinal WAT, gonadal WAT, calf muscle, and trunk blood were collected for each mouse except for one male that was found dead in week 14. Collected peripheral tissues were sliced into multiple pieces and instantly frozen with dry ice.

## Effect of systemic leptin injection and HFD on energy expenditure

Female C57BL/6J mice were used to test the effect of leptin during fasting or HFD on energy expenditure with CLAMS. For the fasting experiment, food was removed from the cage at CT 3.5 hr and leptin (5 mg/kg, i.p.; n = 5) or saline (n = 5) were injected at the same time. Average energy expenditure between the injection time and the end of the light cycle was compared between conditions (*ad lib* fed vs fasting +saline vs fasting +leptin). Energy expenditure after 2 weeks of HFD feeding was measured with female C57Bl/6J mice (n = 5).

## Immunohistochemistry and histological analysis

Brains were sliced at 30 µm thickness into four series with a sliding microtome. pSTAT3 was visualized by diaminobenzene (DAB; 34065, ThermoFisher Scientific) following treatment with Vectastain ABC kit (PK-6100, Vector Laboratories) after incubation with a biotinylated secondary antibody. BAT was formalin fixed, paraffin processed, and sectioned at 5 µm for staining with hematoxylin and eosin (H and E) or with UCP1. UCP1 staining was visualized by DAB using Bond Polymer Refine Detection kit (DS9800, Leica). Stained slides were scanned using a NanoZoomer slide scanner (Hamamatsu) and lipid droplet area was computed on H and E images using a custom app within Visiopharm version 2017.7 (Visiopharm). Primary antibodies used in this study are rabbit anti-pSTAT3 (Tyr705, 9131, Cell Signaling; 1:500) and chicken anti-GFP (ab13970, Abcam; 1:1000), rabbit anti-TRH (Dr. Eduardo A Nillni, Brown University; 1:1000), and rabbit anti-UCP1 (ab209483, Abcam, 1:4000). Secondary antibodies are donkey anti-rabbit IgG-biotin (711-065-152, Jackson ImmunoResearch Laboratories; 1:1000) and donkey anti-chicken IgY-DyLight 488 (703-486-155, Jackson ImmunoResearch Laboratories; 1:200), and donkey anti-rabbit Alexa Fluor 594 (A-21207, ThermoFisher Scientific; 1:200).

## Gene expression analysis

BAT RNA from the POA *Lepr* ablation study with HFD was purified with TRIzol Reagent (15596, ThermoFisher Scientific) and cDNA was synthesized with SuperScript VILO cDNA synthesis kit (11754050, ThermoFisher Scientific). The amount of transcripts for *Ucp1*, *Ppcargc1a*, and *Dio2* were measured using Taqman assays (ThermoFisher Scientific) by real-time PCR (7900HT, ThermoFisher Scientific), and *Gapdh* was used as a reference gene for relative quantification.

## Total plasma T3 measurement

Collected trunk blood from the POA *Lepr* ablation study was immediately mixed with 80 µl of 0.5 M EDTA and then with 15 µl of a protease inhibitor mixture (5 µl of protease inhibitor cocktail (p8340, Millipore Sigma)+5 µl of DPP IV inhibitor (DPP4, Millipore Sigma)+5 µl of 0.5 mg/ml Pefabloc (11429868001, Millipore Sigma)). Blood was centrifuged at 4°C for 10 min at 3000 rpm and then the upper plasma layer was collected and stored at −80°C. The total plasma T3 level was measured with Milliplex Map Kit (RTHYMAG-30K, Millipore Sigma).

Plasma of group two mice (n = 10) from the intra-POA leptin study was similarly prepared and the total T3 was measured the same way.

## POA pSTAT3⁺ cell counting

To determine the level of Lepr signaling in the POA of *Lepr^flox/flox* mice injected with AAV-GFP or AAV-GFP:Cre, the total number of POA pSTAT3$^+$ cells were counted. For each mouse, we pre-selected brain sections that contain rostral, medial, or caudal parts of the median preoptic areas that together encompass bregma 0.62–0.26 mm based on the Paxinos mouse brain atlas (*Paxinos and Franklin, 2004*). For distribution of Lepr neurons in the POA, refer to our previous study (*Yu et al., 2016*). pSTAT3$^+$ cells were counted by the count tool in Adobe Photoshop CS6 (Adobe Systems). Please note that pSTAT3$^+$ cell numbers are only estimates because the POA areas used for cell counting are slightly different between mice in spite of our best effort to minimize the variability. The estimated pSTAT3$^+$ cell numbers in the POA were correlated to various measurements (e.g. body weight).

## Statistical analysis

Data are represented as mean ± SEM. All statistical analyses were done with SPSS 22/24 (IBM) and $p < 0.05$ was considered statistically significant. In all graphs, *$p < 0.05$, **$p < 0.01$, and ***$p < 0.001$. Some bar graphs used letters to indicate statistical significance between comparisons. For more detailed information, see results and figure legends.

## Acknowledgements

This work was supported by AHA053298N, P/F DK020572-30, R01DK092587 (HM), P20GM103528 (HM and SY), 2P30-DK072476 (HM and SY), T32DK064584 (EQC), R01DK101379, R01DK117281, AHA17GRNT32960003, USDA/CRIS3092-5-001-059 (YX), ADA1-17-PDF-138 (YH), R01HL122829 (AVD), R01DK099598 (AZ), R01DK105032 (CM), and R01DK047348 (HRB). This work utilized the facilities of the Cell Biology and Bioimaging Core and Genomics Core supported in part by COBRE (NIH P20GM103528) and CNRU (NIH 1P30DK072476) center grants from the National Institutes of Health. Partial support was provided through the Animal Phenotyping Core supported through NIDDK NORC Center Grant P30 DK072476.

## Additional information

### Funding

| Funder | Grant reference number | Author |
|---|---|---|
| National Institute of Diabetes and Digestive and Kidney Diseases | P30DK072476 | Sangho Yu<br>Yong Xu<br>Andrea Zsombok<br>Christopher D Morrison<br>Hans-Rudolf Berthoud<br>Heike Münzberg |
| National Institute of Diabetes and Digestive and Kidney Diseases | P20GM103528 | Sangho Yu<br>Yong Xu<br>Andrea Zsombok<br>Christopher D Morrison<br>Hans-Rudolf Berthoud<br>Heike Münzberg |
| National Institute of Diabetes and Digestive and Kidney Diseases | R01DK092587 | Sangho Yu<br>Yong Xu<br>Andrea Zsombok<br>Christopher D Morrison<br>Hans-Rudolf Berthoud<br>Heike Münzberg |
| National Institute of Diabetes and Digestive and Kidney Diseases | DK101379 | Sangho Yu<br>Yong Xu<br>Andrea Zsombok<br>Christopher D Morrison<br>Hans-Rudolf Berthoud<br>Heike Münzberg |

| | | |
|---|---|---|
| National Institute of Diabetes and Digestive and Kidney Diseases | 1HL122829 | Sangho Yu<br>Yong Xu<br>Andrea Zsombok<br>Christopher D Morrison<br>Hans-Rudolf Berthoud<br>Heike Münzberg |
| National Institute of Diabetes and Digestive and Kidney Diseases | DK099598 | Sangho Yu<br>Yong Xu<br>Andrea Zsombok<br>Christopher D Morrison<br>Hans-Rudolf Berthoud<br>Heike Münzberg |
| National Institute of Diabetes and Digestive and Kidney Diseases | DK105032 | Sangho Yu<br>Yong Xu<br>Andrea Zsombok<br>Christopher D Morrison<br>Hans-Rudolf Berthoud<br>Heike Münzberg |
| National Institute of Diabetes and Digestive and Kidney Diseases | DK047348 | Sangho Yu<br>Yong Xu<br>Andrea Zsombok<br>Christopher D Morrison<br>Hans-Rudolf Berthoud<br>Heike Münzberg |
| National Institute of Diabetes and Digestive and Kidney Diseases | 1DK117281 | Sangho Yu<br>Yong Xu<br>Andrea Zsombok<br>Christopher D Morrison<br>Hans-Rudolf Berthoud<br>Heike Münzberg |
| ADA Foundation | ADA1-17-PDF-138 | Yanlin He |
| American Heart Association | AHA053298N;<br>AHA17GRNT32960003 | Yong Xu<br>Heike Münzberg |
| U.S. Department of Agriculture | USDA/CRIS3092-5-001-059 | Yong Xu |

The funders had no role in study design, data collection and interpretation, or the decision to submit the work for publication.

## Author contributions

Sangho Yu, Conceptualization, Data curation, Formal analysis, Validation, Investigation, Visualization, Methodology, Writing—original draft, Project administration, Writing—review and editing; Helia Cheng, Investigation, Methodology; Marie François, Clara Huesing, Christopher D Morrison, Hans-Rudolf Berthoud, Visualization, Writing—review and editing; Emily Qualls-Creekmore, Writing—review and editing; Yanlin He, Yanyan Jiang, Hong Gao, Data curation, Formal analysis, Validation, Investigation, Methodology, Writing—review and editing; Yong Xu, Formal analysis, Supervision, Project administration, Writing—review and editing; Andrea Zsombok, Formal analysis, Supervision, Funding acquisition, Validation, Investigation, Visualization, Methodology, Project administration, Writing—review and editing; Andrei V Derbenev, Formal analysis, Supervision, Validation, Investigation, Visualization, Methodology, Project administration, Writing—review and editing; Eduardo A Nillni, Resources, Dr. Nillni has been added as a co-author due to the newly generated data on possible TRH innervation. Dr. Nillni has provided us with a critical antibody resource to stain for TRH cell bodies. This is the only antibody to our knowledge that allows the reliable identification of TRH cell bodies in the PVH; David H Burk, Data curation, Formal analysis, Investigation, Methodology, We have added Dr. Burk as a co-author based on the newly added data on brown adipose tissue histology. Dr. Burk has performed all histological methodologies (cutting, staining, imaging) and has analyzed lipid droplet size; Heike Münzberg, Conceptualization, Resources, Supervision, Funding acquisition, Validation, Visualization, Methodology, Writing—original draft, Project administration, Writing—review and editing

## Author ORCIDs

Sangho Yu http://orcid.org/0000-0002-2973-7562
Yanlin He https://orcid.org/0000-0002-5471-9016
Heike Münzberg http://orcid.org/0000-0002-1152-4884

## Ethics

Animal experimentation: This study was performed in strict accordance with the recommendations in the Guide for the Care and Use of Laboratory Animals of the National Institutes of Health. All of the animals were handled according to approved institutional animal care and use committee (IACUC) protocols (#984; #995) of the Pennington Biomedical Research Center. The protocol was approved by the Institutional Animal Care and Use Committee of the Pennington Biomedical Research Center, which is an AAALAC accredited institution (Animal welfare assurance number: A3677-01). All surgery was performed under deep isoflurane anesthesia, and every effort was made to minimize suffering.

## Decision letter and Author response

Decision letter https://doi.org/10.7554/eLife.33505.022
Author response https://doi.org/10.7554/eLife.33505.023

## Additional files

### Supplementary files

• Transparent reporting form
DOI: https://doi.org/10.7554/eLife.33505.020

### Data availability

Raw electrophysiological data for individually recorded neurons are available in the laboratory of Dr. Xu (Figure 1) and Dr. Zombok (Figure 1—figure supplement 1). Raw metabolic data (food intake and energy expenditure) for individual animals generated and analyzed in Figures 2, 3, 5 are available in the laboratory of the corresponding authors (Drs. Muenzberg and Yu). All histological samples and images generated and analyzed in Figure 2, Figure 2—figure supplement 1, Figure 4—figure supplement 1, Figure 5—supplement 1, Figure 5—supplement 3 and Figure 5—figure supplement 4 are available in the laboratory of the corresponding authors (Drs. Muenzberg and Yu). Raw data generated from T3 ELISAs (Figure 5 and Figure 5—figure supplement 3) and qPCRs (Figure 5—figure supplement 2) are available in the laboratories of the corresponding authors (Dr. Muenzberg and Yu). Raw and numerical data are available with the article as source data files.

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
