## [Decision Letter]

Thank you for submitting your article "Preoptic leptin signaling modulates energy balance independent of body temperature regulation" for consideration by *eLife*. Your article has been favorably evaluated by Mark McCarthy (Senior Editor) and three reviewers, one of whom is a member of our Board of Reviewing Editors. The reviewers have opted to remain anonymous.

The reviewers have discussed the reviews with one another and the Reviewing Editor has drafted this decision to help you prepare a revised submission.

Summary:

Yu and colleagues examined the effects of manipulating leptin signaling in the preoptic area of the hypothalamus on body temperature, energy expenditure and energy balance. They report that preoptic leptin signaling regulates energy expenditure and body weight homeostasis. In contrast, leptin action in the preoptic area did not affect ambient temperature-dependent responses.

Having previously shown a role for POA leptin receptor neurons in thermal sensing and activating these cells decreases food intake and energy expenditure, the authors have now examined the role for direct leptin action on these cells for energy balance. Leptin depolarizes most POA LepRb neurons in acute slice preparations. Direct leptin infusion into the POA briefly decreases food intake, but does not alter energy expenditure (at a variety of temperatures) or energy balance. They also use AAV-cre to delete POA leptin receptors. While this does not alter food intake or energy balance on chow diet, leptin did not increase the energy expenditure when these animals were fasted (this may also have prevented the fall in energy expenditure with fasting). The knockout mice also gained more weight on HFD than did controls. In contrast, no alteration in energy expenditure was detected by calorimetry, the feed efficiency of these mice was increased, suggesting a subtle decrease in energy expenditure on HFD. Collectively, the studies provide some novel observations that will be of interest. However, several issues need to be addressed.

Essential revisions:

The biggest issue is noted by the authors relating to the mechanism of the effects to regulate energy expenditure. The negative data in the BAT is confusing. They suggest it may be related to the thyroid axis, though this correlation is somewhat weak. If the effect is mediated by changes in thyroid activity, some additional experiments supporting this seem to be lacking.

The photomicrographs demonstrating the AAV injections and pSTAT3 are too small for my eyes to see what the authors are presenting. Also, did the authors make maps/drawings of the injection sites and pSTAT staining?

The results are consistent with an effect of POA leptin on energy expenditure, especially when leptin decreases (as with fasting) or increases (as with HFD). The effect size is rather small, however. While the importance of POA leptin action is of some interest, the small effect size and the opposing effects of POA leptin as revealed by injection versus knockout are a source of confusion and limit the impact of the manuscript.

Can the authors be sure that the injected leptin only activates POA LepRb neurons? Mapping pSTAT3 plumes could shed light on this issue.

---

## [Author Response]

Essential revisions:The biggest issue is noted by the authors relating to the mechanism of the effects to regulate energy expenditure. The negative data in the BAT is confusing. They suggest it may be related to the thyroid axis, though this correlation is somewhat weak. If the effect is mediated by changes in thyroid activity, some additional experiments supporting this seem to be lacking.

We agree with the reviewers that the body weight phenotype without a clear effect on energy expenditure is puzzling. However, we would like to remind the reviewers that the body weight changes are mild and small changes in BAT and thyroid function may be below detection levels.

To address this issue, we have now performed additional analysis of BAT histology with H and E staining, lipid area analysis, and UCP1 protein staining. While all parameters trended towards changes that are consistent with decreased thermogenic function in LepRb^POA^ knockdown (KD) mice, these changes did not reach statistical significance. This is consistent with our earlier data showing decreased expression of thermogenic BAT genes in LeprbPOA KD mice, which nevertheless did not reach statistical significance. Our new histological data are now included as Figure 5—figure supplement 3A.

We also performed additional studies to further investigate the link between POA leptin and total T3. Based on the work of Ahima RS et al.1996, we tested whether POA leptin would be sufficient to restore fasting-depressed circulating total T3 levels, using a new cohort of POA cannulated mice (n=10) for infusion of leptin into the POA. We found that POA leptin increased circulating free T3 levels in fasted mice, but these results did not reach statistical significance. We would like to point out that any intraparenchymal cannulation (here in the POA) will introduce more data variation due to cannula placements that explains the high standard error found in the leptin-treated group. These new data are now included in Figure 5—figure supplement 3B.

Finally, we tested whether LepRb^POA^ neurons project to the PVH and are synaptically connected to TRH neurons in the PVH. When we virally expressed farnesylated EGFP in LepRb^POA^ neurons, we observed strong fiber projection of LepRb^POA^ neurons to the PVH in close proximity of TRH neurons (Figure 5—figure supplement 4A-B). We also virally expressed WGA^GFP^, a transsynaptic anterograde tracer, in LepRb^POA^ neurons and found several WGA^GFP^-positive neurons in the PVH, confirming synaptic connectivity of LepRb^POA^ > PVN circuits. However, only a single case of WGA^GFP^-TRH co-expression was found (Figure 5—figure supplement 4C-D), indicating that TRH neurons are likely an indirect target of LepRb^POA^ neurons.

In conclusion, we have now provided several additional and independent data that consistently point towards a blunted BAT thermogenesis and thyroid axis that explain the mild, but significant increase in body weight, but only the direct correlation of POA leptin responsiveness with T3 levels revealed significant changes. We have further highlighted in the Discussion that due to the small non-significant changes, we are unable to pinpoint the exact mechanism for the increased body weight, but that several experimental outcomes suggest a combination of a blunted BAT thermogenesis and thyroid axis (Discussion, fourth paragraph).

The photomicrographs demonstrating the AAV injections and pSTAT3 are too small for my eyes to see what the authors are presenting. Also, did the authors make maps/drawings of the injection sites and pSTAT staining?

We moved previous Figure 4B to Figure 4—figure supplement 1A to enlarge the image. To help visualization of reduced pSTAT3 by AAV-GFP:Cre, we replaced the previous representative histological images for AAV-GFP:Cre to emphasize that the pSTAT3 level decreases only in areas of strong virus expression. We also included a diagram showing virus spreads of all 9 mice injected with AAV-GFP:Cre in Figure 4—figure supplement 1B. We did not include a map of pSTAT3 staining because pSTAT3 shows a complementary/reciprocal pattern to virus spread, as shown in Figure 4—figure supplement 1A.

We similarly enlarged images for AAV injections and pSTAT3 for the HFD study in Figure 5—figure supplement 1A and included virus spread for all 16 mice injected with AAV-GFP:Cre in Figure 5—figure supplement 1B. Previous Figure 5—figure supplement 1B-J are now separated as Figure 5—figure supplement 2.

The results are consistent with an effect of POA leptin on energy expenditure, especially when leptin decreases (as with fasting) or increases (as with HFD). The effect size is rather small, however. While the importance of POA leptin action is of some interest, the small effect size and the opposing effects of POA leptin as revealed by injection versus knockout are a source of confusion and limit the impact of the manuscript.Can the authors be sure that the injected leptin only activates POA LepRb neurons? Mapping pSTAT3 plumes could shed light on this issue.

As shown in Figure 5—figure supplement 1A and in the Discussion, knockdown efficiency of LepRb by AAV-GFP:Cre is only about 50% in the HFD study, which may have contributed to the small effect size.

Observing different effects from POA leptin injection versus knockdown is not entirely surprising because numerous previous studies contrasted effects of low leptin (e.g. fasting, ob/ob) versus high leptin (e.g. IP or ICV leptin injection) on various physiological responses. In other words, leptin-mediated biological responses do not necessarily show a linear correlation to leptin concentration as mentioned in the first paragraph of the Discussion. We did not experimentally address the contrasting phenotypes observed by leptin injection versus knockdown, as this is outside the scope of this manuscript. However, as suggested by the reviewers, we verified that POA leptin injections did not cause leptin leakage to other hypothalamic sites like the ARC. We included representative pSTAT3 images in the POA and the ARC from intra-POA PBS, intra-POA leptin, and IP leptin (Figure 2—figure supplement 1B). We observed very few pSTAT3^+^ cells in the ARC of intra-POA leptin in all animals, thus ruling out leptin effects on the ARC.